# The Proportion of Occupationally Related Cholangiocarcinoma: A Tertiary Hospital Study in Northeastern Thailand

**DOI:** 10.3390/cancers14102386

**Published:** 2022-05-12

**Authors:** Anantapat Seeherunwong, Naesinee Chaiear, Narong Khuntikeo, Chatchai Ekpanyaskul

**Affiliations:** 1Department of Community, Family and Occupational Medicine, Faculty of Medicine, Khon Kaen University, Khon Kaen 40002, Thailand; anantapat.se@kkumail.com; 2Department of Surgery, Faculty of Medicine, Khon Kaen University, Khon Kaen 40002, Thailand; knaron@kku.ac.th; 3Cholangiocarcinoma Screening and Care Program (CASCAP), Faculty of Medicine, Khon Kaen University, Khon Kaen 40002, Thailand; 4Department of Preventive and Social Medicine, Faculty of Medicine, Srinakharinwirot University, Bangkok 10110, Thailand; chatchaie@g.swu.ac.th

**Keywords:** cholangiocarcinoma, occupational exposure, 1,2-dichlropropane, dichloromethane, asbestos

## Abstract

**Simple Summary:**

Northeastern Thailand has the world’s highest incidence rate of cholangiocarcinoma (CCA), whereas a consequence, approximately 14,000 patients die annually. In most cases, the causal factors are identified, but, for some, they remain unknown. Legally imported industrial chemicals such as 1,2-dichloropropane (1,2-DCP), dichloromethane (DCM), and asbestos fibers are defined as occupational causes of CCA. An investigation into these vis-à-vis the diagnosis of occupationally related CCA in Thailand has not been conducted, but is important for understanding the potential magnitude of the problem. The current study found that the proportion of occupationally related CCA was approximately 5.5%, as well as a lower proportion of occupational history taken by treating physicians. Improving physician skills and developing an assistive tool for exploring occupational history might improve the documentation of work-related conditions.

**Abstract:**

Northeastern Thailand registers the highest worldwide incidence of cholangiocarcinoma (CCA). Most of the cases are associated with liver flukes, while unknown causes comprise approximately 10–30% of cases, and these could be due to occupational exposures. Our aim was to determine the magnitude of occupational causes of CCA in a tertiary hospital in northeastern Thailand. We conducted a cross-sectional study with a sample of 220 patients between March and November 2021. Descriptive statistics were used to analyze the findings. Clinical information and telephone interviews were used to explore significant occupational histories. An occupational consensus meeting was held with two occupational physicians, an industrial hygienist, and a hepatobiliary surgeon to decide on the final diagnosis. The response rate was 90.9% (200/220). Based on the medical records and telephone interviews, researchers found that 11 participants had significant exposure. After occupational consensus, it was agreed that the eleven had possible occupational causes, 5.5% (11/200)–54.5% (6/11) being due to asbestos fibers, 45.5% (5/11) due to dichloromethane, and 9.1% (1/11) due to 1,2-dichloropropane. Only 4% (8/200) had occupational histories collected by their treating physicians. Taken together, occupationally related CCA appears to have been underestimated, so improving occupational history taking is needed to properly identify and classify work-related CCA—both for patient treatment and occupational hazard prevention.

## 1. Introduction

Cancer kills 8.2 million people globally, and 14 million cancers are detected every year, according to the WHO International Agency for Research on Cancer (IARC) [1]. The estimated proportion of cancer deaths due to occupational causes is between 4 and 6% and classified by the IARC as a Group 1 (established) or Group 2A (probable) human carcinogen for a specific cancer site [2,3]. Presently, several research and development studies are addressing various aspects of CCA treatment [4,5,6].

The incidence rate for cholangiocarcinoma (CCA) varies substantially worldwide, reflecting local geographic risk factors [7]. Northeastern Thailand continues to have the highest global incidence of CCA, resulting in approximately 14,000 deaths annually [8]. The causes of CCA are multifactorial, but significant risk factors include liver fluke infestation, past illness (viz., primary sclerosing cholangitis, fibrocystic liver disease, hepatitis B and C, and biliary stones), and thorotrastic exposure [9]. Apart from these known risk factors, another 10–30% of cases have unknown causes [10,11], some of which may be occupationally related.

In 2012, a cluster of 17 CCA patients was reported by a small Japanese company. All of the patients were exposed to long-term, high levels of inhaled and dermal dichloromethane (DCM) and 1,2-dichloropropane (1,2-DCP) at an ink removal operation [12]. Animal studies have revealed the mechanism by which DCM and 1,2-DCP influence glutathione-S-transferase theta 1 (GSTT1), thereby contributing to cholangiocarcinogenesis [13,14]. The IARC thus classified the 1,2-DCP and DCM and as classes 1 and 2A carcinogen, respectively [15]. Not only is DCM used in the printing industry, but it is also used in numerous industrial settings, such as paint stripping, pharmaceuticals, paint remover manufacture, degreasing, and metal cleaning agents [16].

Asbestos fibers have also been implicated in CCA development [17,18]. The deposition of asbestos fibers in the biliary tract was confirmed in previous studies [19,20]. The researchers outlined how the ingestion/inhalation of asbestos fibers can lead to fibers being trapped in the small bile ducts, inducing chronic inflammation and being followed by impaired cell proliferation, resulting in malignancy [21,22,23].

The Thai Customs Department reported that the importation of these chemical continues, and yet, reports of the diagnosis of occupationally related CCA are not being published. A better estimate of the magnitude of the problem is needed, as well as clearly defining risky jobs, which would help in establishing proper prevention strategies for exposed employees. The current study thus aimed to evaluate the proportion of possible occupationally related causes among CCA patients at a tertiary hospital in northeastern Thailand.

## 2. Materials and Methods

### 2.1. Study Design

We conducted a cross-sectional descriptive study at a tertiary hospital in northeastern Thailand.

### 2.2. Study Population and Sample

The study involved patients diagnosed with ICD-10 codes C22.0 (malignant neoplasm of liver and intrahepatic bile duct), C24.0 (extrahepatic bile duct malignant neoplasm), or SNOMED CT 312104005 (CCA of biliary tract). All participants were without any underlying condition with a high risk for CCA (viz., primary sclerosing cholangitis, fibro polycystic liver disease, cirrhosis, hepatitis B or C, and/or biliary stones) [9] and no family history of CCA [24]. Data were collected between March and November 2021. The target population of 293 patients was drawn from a tertiary hospital in northeastern Thailand. The included participants (a) had a diagnosis of CCA and (b) were younger than or equal to 80 years old. The exclusion criteria were the inability to communicate due to medical reasons (e.g., presbycusis or psychiatric conditions.) The final study population comprised 220 persons. The sample size was calculated using G-Power version 3.1.9.3 (HHU., Düsseldorf, Germany). The formula was based on estimating a proportion with the following indicators: a confidence level of 95%, a precision of 0.02, an assumed 20% proportion of occupational CCA due to unknown risk factors [10,11], and a population of 293 participants. The required sample size was thus 220 participants, so all 220 eligible persons in the study population were included in the sample. Twenty persons did not respond—sixteen did not answer the call, three gave incorrect phone numbers, and one refused to join the research (Figure 1).

### 2.3. Tool and Data Collection

After ethics approval, we sent a letter to the hospital director requesting permission to access the electronic database and paper records. Clarification forms were sent to each potential participant at the surgery clinic on the appointed day for joining the study and requesting their call-back information. An occupational medicine resident interviewed the participants by telephone as per the arranged schedule. The resident was trained and experienced in occupational history taking. The interview data underwent content validation by an experienced occupational physician. If a participant did not answer the call, they were reclassified into the non-respondent group.

The telephone interview was used to collect the following data: underlying disease, smoking and drinking status, liver fluke infection, thorotrastic exposure, history of CCA in a co-worker, job(s) held, job description(s), working status, age started working, retirement age, and occupational exposure. In addition, the latter included related processes, frequency of exposure, and duration of exposure (Appendix A).

Clinical information was gained from hospital medical records. The following data were reviewed: (a) demographics (age, age at diagnosis, sex); (b) type of CCA (intrahepatic, extrahepatic); and, (c) occupational history recorded by treating physicians.

If the participant was likely to be exposed, but could not remember precisely when, we involved an industrial hygienist to help confirm the chemical exposure. An occupational consensus meeting was then convened to confirm the occupationally related CCA diagnosis.

### 2.4. Multidiscipinary Team Consensus

In the current study, since we were unable to calculate cumulative exposure on an individual basis, researchers defined significant occupational exposure if a worker used personal protective equipment (PPE) inappropriately coincidental with history of: (1) exposure to DCM or 1,2-DCP of more than 6.1 work years (estimated 12,520 h) [25]; and (2) exposure to asbestos at work for more than 30 consecutive days per year for 30 years [18]. Participants who had significant occupational exposure were selected for an occupational consensus discussion with two occupational physicians, an industrial hygienist, and a hepatobiliary surgeon. Clinical information retrieved for the meeting included age, sex, underlying disease, and type of CCA. Other information explored during the telephone interview included lifelong occupational history, history of liver fluke infection, thorotrastic exposure, and history of CCA. The collected information was evaluated to confirm each diagnosis, which was categorized into three groups depending on the degree of confidence: “Definite” if the diagnostic confidence was ≥90% with a compatible history and laboratory test; “Possible” if the diagnostic confidence was 50–89% with a compatible history but some risk factors being uncertain; and, “unclassifiable” if the diagnostic confidence was <50%.

### 2.5. Data Analysis

Descriptive statistics were used to analyze the data, including frequencies, percentages of possible work-related CCA, and the 95% confidence interval (95%CI). All analyses were performed using SPSS version 29.0 (IBM SPSS Inc, Chicago, IL, USA).

## 3. Results

### 3.1. General Characteristic of Participants (n = 200)

Most of the eligible study population responded to the telephone interview (90.9%; 200/220). Most participants were male (62.5%). The median age was 62 years (IQ.R = 49–75; range 39–80). Hypertension was the most common underlying disease (24.5%), followed by diabetes (18.5%) and dyslipidemia (9%). Most of the participants had a history of eating raw fish (83.5%; 167/200) and using anthelmintics (77.5%; 155/200). Only 14.5% had a history of liver fluke infection. Almost three-quarters were diagnosed between 51 and 70 years of age. Almost half of the participants never smoked, but more than half had a history of moderate-heavy alcohol consumption. None of the participants had a history of thorotrastic exposure (Table 1).

Information on occupational history was available for only 4% of the participants (8/200). Typically, the work history recorded only one job title. Almost half of the participants had intrahepatic CCA. Most participants lived in northeastern Thailand, and none were from southern Thailand (Table 2).

### 3.2. Occupational Characteristic (n = 200)

One fourth of the participants reported that they were presently working. The estimated age for starting work was a median of 16 years (IQR = 14–20; range 12–25). Most had more than one occupation (64.5%). Extra jobs were regularly related to construction. The longest held occupation was agriculture-related. According to the interview, 11 participants had a history of significant occupational chemical exposure (5.5%; 11/200), where the majority (54.5%; 6/11) were exposed to asbestos fibers, whereas 45.5% (5/11) were exposed to dichloromethane and 1,2-dichloropropane 9.1% (1/11) (Table 2).

### 3.3. Work-Related CCA (n = 11)

According to the occupational team discussion, all of the participants with a significant exposure history were included in the occupational-related CCA. The majority experienced asbestos exposure as roof workers, whereas those with DCM and 1,2-DCP exposure had various occupations (including printing worker, mechanic, worker at film factory, vocational teacher, and welder). All 11 participants were male, 9 had smoked and 8 had consumed alcohol, and more than half had intrahepatic CCA, which was particularly prevalent among those with asbestos exposure at 83.3% (5/6).

The mean age was 61.8 (SD = 4.1; range 57–68) among those with DCM or 1,2-DCP exposure, followed by 59.3 (SD = 4.5; range 55–65) for those with asbestos exposure. The mean duration of exposure was 19.4 years (SD = 15.1, range = 10–40) and 35.5 years (SD = 4.5, range 30–43), respectively. Participants with significant exposure to DCM or 1,2-DCP developed into CCA with a latency period of around 27 years (mean = 27.2, SD = 13.2; range 10–41) followed by 36.5 (SD = 3.6; range 34–43) for those with asbestos exposure (Table 3 and Table 4).

## 4. Discussion

The current study is the first to clarify the proportion of occupationally related CCA. Until recently, the diagnosis of occupationally related CCA was not being reported, even though 1,2-DCP, DCM, and asbestos are known carcinogens legally imported into Thailand. The current study revealed the existence of occupationally related CCA: A finding useful for policy development related to occupational health and for reconsidering the importation of these chemicals.

As for occupationally related CCA, there is no standard or previous study setting forth the criteria for diagnosis, although the literature does include a cluster of occupationally related CCA among workers at a printing company in Japan [12]. Thus, an occupational consensus discussion was held to determine a work-related diagnosis. The team included industrial hygienist, hepatobiliary surgeon, and occupational doctor as per Horlait et al., who suggested that occupational data were necessary for a multidisciplinary team meeting in cancer care to improve the quality of care [26]. Some studies suggest having a pathologist or radiologist on the team [27], but our study did not include such persons because all of our participants already had tissue-confirmed CCA by a pathologist.

The study found only 5.5% of 200 CCA diagnosed in a tertiary hospital in northeastern Thailand (95% CI; 2.3–8.7), which is within the range of previous studies that have estimated 4–6% of cancer deaths due to occupational causes [2,3,28] that could be occupationally related. Precisely, these patients had no other risk factors, including primary sclerosing cholangitis, fibro polycystic liver disease, hepatitis B or C viral infection, cirrhosis, biliary stones, history of CCA in family, or thorotrastic exposure. It is difficult to confirm which other risk factor was more causative of occupationally related CCA, since they have similar strengths of association vis-à-vis CCA [9]. Importantly, cigarette smoking and alcohol consumption were not excluded in the current study, although previous studies found a weak/modest association with CCA (OR: 1–1.7). By comparison, asbestos exposure has a strong association (OR > 3) and 1,2-DCP exposure has a very strong association (OR > 8) [9]. During occupational consensus, it was agreed that cigarette smoking and alcohol consumption did not represent a significant risk factor for occupationally related CCA. In particular, one of the cases could be related to DCM and/or 1,2-DCP exposure. The period of chemical exposure prior to diagnosis was 12,480–29,120 working hours (6–14 working years) as per a prior study wherein multiple workers had developed CCA ranging from 6 years and 1 month to 16 years (median 12 years, 6 months) [25]. Participants recently diagnosed with CCA after 14 years of exposure could have occupationally related CCA despite it being 18 years since the end of exposure [29]. Four of the participants were exposed only to DCM without 1,2-DCP.

The relationship between DCM and CCA has been controversial and there is limited evidence on their carcinogenicity to humans (Group 2A) [15]. However, at least two studies have found a relationship with biliary tract cancer in industries where DCM is regularly used [30,31,32]. The mechanism of carcinogenesis is that DCM proceeds through the glutathione-S-transferase theta 1 (GSTT1) catalytic pathway, resulting in the production of reactive intermediates implicated in cholangiocarcinogenesis [15]. In addition, some experimental studies have revealed the influence of glutathione S-transferase theta1 (GSTT1) in DCM exposure causing mutagenicity, with preferential expression in bile duct epithelial cells rather than hepatocytes. The mechanisms of occupational CCA development, but not hepatocellular carcinoma, are not fully understood [33,34,35].

Most of the participants (6/11) had an extensive history of asbestos exposure during the manufacture or demolition of asbestos-containing products (i.e., roof tile and cement pipe) [36]. Other studies similarly identified an association between asbestos and CCA [17,18]. Exposure can result in asbestos fibers in the bile ducts: (a) fibers can be drained by convective flow into initial pulmonary lymphatic and potentially translocated to all organs via down pressure gradients [37]; (b) large fibers can be trapped in the smaller bile ducts by the high microvascular permeability of the liver sinusoid [22,38]; and (c) fibers in the liver can give rise to a chronic inflammatory status. In addition, the production of oxygen radicals, cytokines, and growth factors can impair cell proliferation and apoptosis, resulting in DNA damage leading to malignant transformation [23].

This study aimed to evaluate the magnitude of the occupationally related causes in participants who had no other main risk factors. Since 1,2-DCP, DCM, and asbestos could be the trigger mechanisms for malignant transformation by the chronic inflammatory process, the workers exposed to these chemicals could develop occupational CCA without other risk factors. As for other co-overlapping risk factors, the presence of two or more predicted a higher risk for occupationally related CCA. A previous study showed that DCM co-exposure potentially enhanced 1,2-DCP liver toxicity, resulting in a higher percentage of occupationally related CCA [14]. Notwithstanding, none of the other studies have found an interaction with other risk factors. Consequently, future studies should focus on risk factors that might have some synergic effects on bile duct epithelial damage, leading to malignant transformation, such as cholangitis, polycystic liver disease, and hepatitis infection.

The proportion of occupationally related CCA was underestimated because a lower rate of occupational history was taken by the physician. The current study demonstrated that only 4% (8/200) of the participants had an occupational history taken by their treating physician, and the proportion in previous studies showed that only 24–28% had completed an occupational history [39,40]. The lower rate in this study could be explained by the fact that previous studies not only investigated CCA but also other diseases, which might have impacted the result, indicated by the lower rate. This led to the conclusion that occupational history taking in CCA was not being carried out. In general, physicians might lack awareness of occupationally related conditions or might not be trained in taking occupational history [41]. Improving physician skills and developing an assistive tool for exploring occupational history might improve the documentation of work-related conditions.

The study had some limitations: (a) a lack of a control to confirm the strength of the relationship between exposed and non-exposed participants; (b) a small sample size that does not allow for the generalization of findings; (c) the underestimation of occupationaly related CCA due to incomplete recall inherent in retrospective assessments; (d) the history of liver fluke infestation is a very strong risk factor for CCA, but data from medical records in this study obtained from feces and urine might not provide a precise diagnosis (a more extensive study applying IgG antibodies to *Opisthorchis viverrini* in serum [42] should be conducted to assess the real impact of occupationally related CCA); and (e) a lack of information on the dose and duration of cigarette smoking and alcohol consumption needs more careful consideration.

## 5. Conclusions

The proportion of potentially occupationally related disease among the 200 CCA patients found at a tertiary hospital in northeastern Thailand was 5.5%, which is within the range of the estimated 4–6% of cancer due to occupational causes as per the IARC Group 1 or Group 2A human carcinogens for a specific cancer site [2,28]. In order to determine the precise occupationally related diagnosis, exposure investigations should include work histories. In addition, workers exposed to 1,2-DCP, DCM, or asbestos need long-term health monitoring to ensure the early detection of CCA and timely treatment. As this study had a small sample size, confirmatory research is needed on this emerging issue.

## Figures and Tables

**Figure 1 cancers-14-02386-f001:**
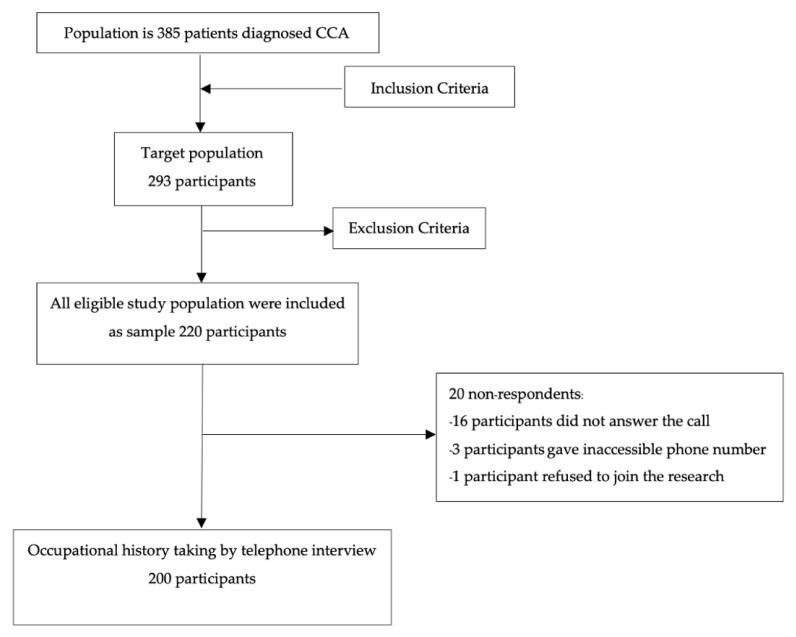
Flow chart of study population and sampling process.

**Table 1 cancers-14-02386-t001:** General characteristics of participants (*n* = 200).

Characteristic	*n*	%
Sex		
Male	125	62.5
Female	75	37.5
Age at diagnosis		
30–40 years	4	2.0
41–50 years	18	9.0
51–60 years	66	33.0
61–70 years	84	42.0
71–80 years	28	14.0
Medical condition		
Hypertension	49	24.5
Diabetes mellitus	39	18.5
Dyslipidemia	18	9.0
Chronic kidney disease	6	3.0
Other	21	10.5
Eating raw fish		
Yes	167	83.5
No	33	17.5
Using anthelmintics		
Yes	155	77.5
No	45	22.5
Liver fluke infection		
Found in laboratory examination	29	14.5
Not found in laboratory examination	71	30.5
Never tested in laboratory examination	78	39.0
Do not remember	22	11.0
Cigarette smoking		
Current smoker	82	41.0
Never smoked	118	59.0
Alcohol consumption		
Current drinker	107	53.5
Never drank	93	46.5
Mean age (IQR)	62 (49–75)	

**Table 2 cancers-14-02386-t002:** Data obtained from medical record and telephone interview (*n* = 200).

Item	*n*	%
Occupational history recorded by treating physicians		
One job title	7	3.5
More than one job tile	1	0.5
Not available	192	96
Type of cholangiocarcinoma		
Intrahepatic	93	46.5
Extrahepatic	107	53.5
Region of Thailand		
Northeast	193	96.5
Central	3	1.5
West	2	1
East	1	0.5
North	1	0.5
Work status		
Working	54	27
Retired	146	73
Occupation (s)		
One	71	35.5
Two	107	53.5
Three	22	11
History of cholangiocarcinoma in co-worker		
Yes	8	4
No	192	96
Longest-held occupation classified by major group of ISCO-68		
Group 0/1: professional, technical, and related workers	16	8
Group 2: administrative and managerial workers	5	2.5
Group 3: clerical and related workers	5	2.5
Group 4: sale workers	9	4.5
Group 5: service workers	12	6
Group 6: agricultural, animal husbandry and forestry workers, fishermen and hunters	122	61
Group 7/8/9: production and related workers,	28	14
Transport equipment operators and labourers		
X: workers not classifiable by occupation	0	0
Y: members of the armed forces	3	1.5
Significant exposure history (*n* = 11)		
Asbestos	6	54.5
Dichloromethane	5	45.5
1,2-Dichloropropane	1	9.1
Mean age starting work (IQR)	16 (14–20)	
Mean years of the longest-held occupation (SD)	34.5 (10.1)	

**Table 3 cancers-14-02386-t003:** Characteristics of participants exposed to 1,2-DCP and DCM diagnosed as possible occupationally related CCA (*n* = 5).

ID	Sex	Age	Possible Job	Job Description	Years of Exposure	Latency Period (Years)	Liver Fluke Infection	Type of CCA
1	male	57	printing worker	ink removal operation by using 1,2 DCP and DCM 4–6 h a day (15,600–18,200 working hours)	10	24	NA	ICC
2	male	68	mechanic	using solvent with mainly DCM to paint removal and stripping trucks 4–6 h a day (12,480–18,720 working hours)	10	10	NA	ECC
3	male	63	worker at film factory	cellulose triacetate film production using DCM in fiber manufacturing 8 h a day (14,976 working hours)	6	21	not found in feces	ECC
4	male	61	vocational teacher	using solvent with mainly DCM to paint removal wooden and metal furniture 2 h a day (16,120 working hours)	31	41	not found in feces	ICC
5	male	60	welder	stripping metal and re-painted by using DCM 1–2 h a day (14,560–29,120 working hours)	40	40	NA	ECC
Mean age (SD)				61.8 (4.1)				
Mean duration of exposure (SD)				19.4 (15.1)				
Latency period (SD)				27.2 (13.2)				

ID: indentification number, CCA: cholangiocarcinoma, ICC: intrahepatic cholangiocarcinoma, ECC: extrahepatic cholangiocarcinoma, NA: not applicable, working hours represent duration of exposure to 1,2-DCP and DCM.

**Table 4 cancers-14-02386-t004:** Characteristics of participants exposed to asbestos diagnosed as possible occupationally related CCA (*n* = 6).

ID	Sex	Age	Possible Job	Job Description	Years of Exposure	Latency Period (Years)	Liver Fluke Infection	Type of CCA
1	male	55	roofing worker	roofing and roof demolition 7–8 h a day (49,504–70,720 working hours)	34	34	Not found in feces	ICC
2	male	63	road worker	cutting, drilling cement sheets with certain asbestos products in asphalt 7–8 h a day (69,160–79,040 working hours)	38	38	N/A	ICC
3	male	65	roofing worker	roofing and roof demolition 7–8 h a day (84,084–96,096 working hours)	33	33	N/A	ECC
4	male	56	construction worker	cut, drilled, sanded, and shaped several asbestos-based building products 1–2 h a day (4,680–12,480 working hours)	30	36	N/A	ICC
5	male	62	construction worker	demolish buildings, roofing, flooring and tiling 7–8 h per day (11,180–18,200 working hours)	35	35	Not found in feces	ICC
6	male	55	roofing worker	roofing and roof demolition 7–8 h a day (15,652–35,776 working hours)	43	43	N/A	ICC
Mean age (SD)				59.3 (4.5)				
Mean duration of exposure (SD)				35.5 (4.5)				
Latency period (SD)				36.5 (3.6)				

ID: indentification number, CCA: cholangiocarcinoma, ICC: intrahepatic cholangiocarcinoma, ECC: extrahepatic cholangiocarcinoma, NA: not applicable, working hours represent duration of exposure to asbestos.

## Data Availability

The data presented in this study are available on request from the corresponding author. The data are not publicly available as the included patients did not specifically provide consent for public sharing of their data and post hoc anonymization was unfeasible.

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
