# Peer review of "The Proportion of Occupationally Related Cholangiocarcinoma: A Tertiary Hospital Study in Northeastern Thailand"

_cancers, 2022, doi:10.3390/cancers14102386_

Round 1

Reviewer 1 Report

Dear Editor, thank you so much for inviting me to revise this manuscript about cholangiocarcinoma.

This study addresses a current topic.

The manuscript is quite well written and organized. English could be improved.

Figures and tables are comprehensive and clear.

The introduction explains in a clear and coherent manner the background of this study.

We suggest the following modifications:

  • Introduction section: although the authors correctly included important papers in this setting, we believe some recent papers should be cited within the introduction ( PMID: 33571059 ; PMID: 33592561; PMID: 32806956 ), only for a matter of consistency. We think it might be useful to introduce the topic of this interesting study.
  • Methods and Statistical Analysis: nothing to add.
  • Discussion section: Very interesting and timely discussion. Of note, the authors should expand the Discussion section, including a more personal perspective to reflect on. For example, they could answer the following questions – in order to facilitate the understanding of this complex topic for readers: what potential does this study hold? What are the knowledge gaps and how do researchers tackle them? How do you see this area unfolding in the next 5 years? We think it would be extremely interesting for the readers.

However, we think the authors should be acknowledged for their work. In fact, they correctly addressed an important topic, the methods sound good and their discussion is well balanced.

One additional little flaw: the authors could better explain the limitations of their work, in the last part of the Discussion.

We believe this article is suitable for publication in the journal although some revisions are needed. The main strengths of this paper are that it addresses an interesting and very timely question and provides a clear answer, with some limitations.

We suggest a linguistic revision and the addition of some references for a matter of consistency. Moreover, the authors should better clarify some points.

Author Response

Dear Reviewer #1

Thank you very much for your comment and suggestion. The response and the revised version are attached. Any further actions, we will be pleased to take it further.

Sincerely yours,

Naesinee Chaiear

Reviewer 2 Report

The authors investigated the proportion of possible occupationally-related causes among CCA patients at a tertiary hospital in northeastern Thailand. The authors found that the proportion of possibly occupationally-related disease among 200 CCA patients at a tertiary hospital in northeastern Thailand was 5.5%. Very interesting findings are made; however, my comments are as follows:

.1 “The causes of CCA are multifactorial, but significant risk factors include liver fluke infestation, past  llness (viz., primary sclerosing cholangitis, fibrocystic liver disease, hepatitis B and C, and biliary  tones), and thorotrastic exposure [6]. Apart from these known risk factors, there are another 10-30% of cases from other causes [7,8]”. It would be very interesting, for authors to describe briefly, the 10-30% causes of CCA in the introduction.

2. “We conducted a descriptive study at a tertiary hospital in northeastern Thailand.” Did the study adopt a cross-sectional design? This must be mentioned.

3. Combine Tables 2 and 3.

4. Under Tables 4 and 5 there is a description of working hours e.g. 49,504-70,720. What do these hours represent? Kindly provide a footnote for this.

5. “The lower rate in our study might be a lack of awareness of occupationally-related conditions or that physicians were not trained in taking occupational history [39]”. How true is this statement since Under 2.4 it is indicated that an occupational consensus discussion was held? This could have been spotted during the consensus discussions and accounted for under 2.4. 

Author Response

Thank you very much for your comment and suggestion. The response and the revised version are attached. Any further actions, we will be pleased to take it further.
Please see the attachment.
Sincerely yours,
Naesinee Chaiear

Reviewer 3 Report

This is a really original and interesting paper, dealing with a neglected but emerging issue.

I only suggest to the Authors to:

  • add to the limitations (Discussion-line 266) also the small sample size that doesn't allow to make any generalization;
  • add to the conclusions (line 279) the fact that, given the small sample size, the present study gives only suggestive and preliminary results, suggesting a need for further research on this emerging issue. 

Author Response

(The authors gave the same response as above.)

Round 2

Reviewer 1 Report

The authors addressed all our queries.

We recommend Acceptance in its current form.

Author Response

Thank you so much for your support. I did not receiv any further comment from the reviwer#1.

Sincerely yours,

Naesinee